# BAG OF FEATURES: NEW BASELINES FOR GNNS FOR LINK PREDICTION

## ABSTRACT

Graph Neural Networks (GNNs) have brought a significant transformation in the realm of graph representation learning. They achieve this by employing a neighborhood aggregation approach, wherein a node's representation vector is iteratively calculated by aggregating and modifying the corresponding vectors of its neighboring nodes. Despite GNNs demonstrating superior performance in various domains over the last ten years, recent theoretical studies have raised concerns about their expressive capabilities, where they show that GNN models yield results comparable to the well-established Weisfeiler-Lehman algorithm.

In this paper, driven by this motivation, we compare the performance of current GNN models with conventional feature extraction methods in the context of link prediction. Our experiments reveal that when applied to standard feature sets derived from node neighborhoods and node features, standard machine learning (ML) models deliver highly competitive results, even when pitted against cutting-edge GNN models. This holds true across both small and large benchmark datasets, including those from the Open Graph Benchmark (OGB). Our empirical findings corroborate the previously mentioned theoretical observations and imply that there exists ample room for enhancement in current GNN models to reach their potential.

## 1 INTRODUCTION

In a time characterized by the intricate web of digital connections, understanding and predicting the formation of links between entities in complex networks has become a crucial challenge. Whether it's predicting social connections in online social networks, anticipating collaborations between researchers, or forecasting potential interactions in recommendation systems, link prediction has emerged as a fundamental task in the realm of graph representation learning. With the ongoing evolution of our societies and technologies, the significance of link prediction amplifies, considering its capability to refine a broad spectrum of applications—ranging from personalized content suggestions to strategic targeted marketing.

Over the past decade, Graph Neural Networks (GNNs) have achieved a significant breakthrough in the field of graph representation learning, in particular link prediction tasks. They have accomplished this by adapting various deep learning models originally developed for diverse domains such as computer vision and natural language processing to the context of graphs (Section 2.1). GNNs employ a variety of approaches and architectural designs to generate robust node embeddings, seamlessly blending neighborhood information with domain-specific features. Despite their consistently superior performance compared to state-of-the-art results, seminal papers like (Xu et al., 2019; Morris et al., 2019; Li & Leskovec, 2022) have shown that the expressive capacity of message-passing GNN models is not better than decades-old Weisfeiler-Lehman algorithm. These theoretical discoveries suggest that existing GNN models may not be fully leveraging their potential to integrate information from local neighborhoods and domain-specific knowledge when learning node embeddings, a fundamental process in the creation of node representations.

In this paper, motivated by these findings, we hypothesize that current GNN models may not offer significant improvements over traditional feature engineering models in link prediction task. In order to empirically validate this hypothesis, we have devised a simple machine learning model by employing conventional feature extraction techniques for link prediction task, *Bag of Features*. For each pair of nodes, we initially derive *structural features* by using their proximity within the graph

structure. Additionally, we acquire *domain features* by evaluating their similarity within the feature space. The fusion of these spatial and domain features, when integrated with standard machine learning methodologies, yields remarkably competitive results when compared to state-of-the-art Graph Neural Network (GNN) models. These outcomes are consistently observed across a diverse range of datasets.

**Our contributions:**

◇ We propose a computationally feasible and scalable ML model *Bag of Features* (BFLP), by adeptly merging existing feature extraction methods for link prediction.

◇ Our model effectively combines the local neighborhood information and domain-specific node features. These extracted features can be seamlessly integrated into any future ML model to enhance their performance.

◇ Our model consistently surpasses or achieves highly competitive results when compared to state-of-the-art GNN models in benchmark datasets.

◇ Our empirical results provide substantial support for recent theoretical findings, which imply that current GNNs may not fully exploit their potential, especially when compared to their counterparts in computer vision and NLP. This underscores the need for innovative approaches to unlock the full capabilities of GNNs.

## 2 RELATED WORK

### 2.1 GRAPH NEURAL NETWORKS FOR LINK PREDICTION

Much of the recent work on link prediction has been using GNNs. Most GNNs follow the message-passing framework, in which a node's representation is learned through an aggregation operation, to pool local neighborhood features, and an update operation, which is a learned transformation (Guo et al., 2023). Another common framework is the encoder-decoder framework in which the encoder learns node representations and the decoder predicts the probability of a link between two nodes (Guo et al., 2023). There are four main groups of GNNs currently used (Wu et al., 2020): recurrent GNNs (RecGNN) (Dai et al., 2018), convolutional GNNs (ConvGNNs) (Chiang et al., 2019), graph autoencoders (GAEs) (Bojchevski et al., 2018), and spatial–temporal GNNs (STGNNs) (Guo et al., 2019).

In the link prediction task, GNNs have shown outstanding performance in the past decade (Zhang, 2022; Zhu et al., 2021; Wang et al., 2023). Zhao et al. (2022) use counterfactual links as a data-augmentation method to obtain robust and high-performing GNN models. In (Yun et al., 2021), the authors applied novel approaches to improve learning structural information from graphs. In (Zhu et al., 2021), the authors integrate the Bellman-Ford algorithm for path representations to their GNN model and obtain competitive results in both inductive and transductive settings. Wu et al. (2021) proposed an effective similarity computation method by employing a hashing technique to boost the performance of GNNs in link prediction tasks. Liu et al. (2022) developed high-performing GNNs for dynamic interaction graph setting. There is an overwhelming literature on GNNs for link prediction in the past few years, and a good review of these results can be found in (Zhang, 2022; Liu et al., 2023; Wu et al., 2022).

### 2.2 FEATURE ENGINEERING FOR LINK PREDICTION

Before the GNNs, many of the common machine learning models often relied on feature engineering methods as a primary approach (Kumar et al., 2020; Menon & Elkan, 2011). One of the simplest approaches to link prediction is through similarity-based methods (Lü & Zhou, 2011), including local similarity indices (Wu et al., 2016), global similarity indices (Jeh & Widom, 2002), and quasi-local indices (Liu & Lü, 2010). In this case, the graphs are mostly assumed to be homophilic, and more similar nodes are deemed more likely to have a link.

Most of the former feature extraction methods for link prediction can be categorized as local similarity indices. Let $S(u, v)$ denote a similarity score between two nodes $u$ and $v$, let $\mathcal{N}(u)$ denote the set of neighbors of a node $u$, and let $k_u$ denote the degree of a node $u$.

*Common Neighbors* is the size of the intersection between two nodes' neighbors (Newman, 2001). This is equivalent to the number of paths of length 2 between two nodes. More common neighbors indicates a higher likelihood for a link.

$$\mathcal{CN}(u,v) = |\mathcal{N}(u) \cap \mathcal{N}(v)|$$

*Jaccard Coefficient* is a normalized Common Neighbor score (Jaccard, 1901). It corresponds to the probability of selecting a common neighbor of two nodes from all neighbors of those nodes.

$$\mathcal{J}(u,v) = \frac{|\mathcal{N}(u) \cap \mathcal{N}(v)|}{|\mathcal{N}(u) \cup \mathcal{N}(v)|}$$

*Salton Index* (also Cosine similarity) measures similarity using orientation rather than magnitude (Singhal et al., 2001).

$$\mathcal{S}_a(u,v) = \frac{|\mathcal{N}(u) \cap \mathcal{N}(v)|}{\sqrt{k_u k_v}}$$

*Sorensen Index* was developed for ecological data samples (Sørensen, 1948), and it is more robust than Jaccard against outliers (McCune & Grace, 2002).

$$\mathcal{S}_o(u,v) = \frac{2|\mathcal{N}(u) \cap \mathcal{N}(v)|}{k_u + k_v}$$

*Adamic Adar Index* basically measures the amount of shared links between two nodes (Adamic & Adar, 2003). It is defined as

$$\mathcal{AA}(u,v) = \sum_{z \in \mathcal{N}(u) \cap \mathcal{N}(v)} \frac{1}{\log |k(z)|}$$

In this paper, we effectively use these traditional similarity indices as our *structural features* for node pairs. We then effectively combine them with our *domain features*, which is the relevant similarity measure between the node features, to complete our feature set for our ML models.

## 3 METHODOLOGY

### 3.1 PROBLEM STATEMENT

Link prediction problems in graph representation learning can be categorized into different types based on the nature of the problem and the availability of information during the prediction process. Three common types of link prediction problems are *transductive*, *inductive*, and *semi-inductive* link prediction.

Let $\mathcal{G} = (\mathcal{V}, \mathcal{E}, \mathcal{X})$ be a graph, where $\mathcal{V} = \{v_1, v_2, \ldots, v_n\}$ represents the set of vertices (or nodes), $\mathcal{E} \subset \mathcal{V} \times \mathcal{V}$ represents the set of edges (or links) and $\mathcal{X}$ represents the attribute feature matrix ($n \times m$ size) where $\mathbf{X}_i \in \mathbb{R}^m$ is the $m$-dimensional attribute feature vector of node $v_i$. For the sake of simplicity, we focus on unweighted, undirected graphs, but our setup can easily be adapted to more general settings.

The main difference between these types comes from the availability of the information during the prediction process. We split the vertex and edge sets as *observed* (old) and *unobserved* (new) subsets, i.e. $\mathcal{V} = \mathcal{V}_o \cup \mathcal{V}_u$ and $\mathcal{E} = \mathcal{E}_o \cup \mathcal{E}_u$. Hence, in the training process, we are provided $\mathcal{G}_o = (\mathcal{V}_o, \mathcal{E}_o)$ information, and we are asked to predict the existence of a link in $\mathcal{E}_u$ for a given node pair in $\mathcal{V}$. However, the type of the problem is determined with respect to which subsets (i.e., $\mathcal{V}_o$ or $\mathcal{V}_u$) these node pairs are chosen from:

- *Transductive Setting:* Predict whether $e_{ij} \in \mathcal{E}_u$ where $v_i, v_j \in \mathcal{V}_o$.
- *Semi-inductive Setting:* Predict whether $e_{ij} \in \mathcal{E}_u$ where $v_i, v_j \in \mathcal{V}_o \cup \mathcal{V}_u$.
- *Inductive Setting:* Predict whether $e_{ij} \in \mathcal{E}_u$ where $v_i, v_j \in \mathcal{V}_u$. No local structure information is provided, only attribute vectors $\{\mathbf{X}_i\}$ are provided for $v_i \in \mathcal{V}_u$.

While, in the literature, the most common type is transductive setting, depending on the domain, the relevant question can come in any of these forms. To maintain focus in this paper, we align with the prevalent transductive setting, consistent with most contemporary GNN models. It is important to note, however, that our proposed feature engineering ML model exhibits a high degree of versatility and can easily adapt to any of these settings.

## 3.2 BFLP: Bag of Features for Link Prediction

In the following, we give the details of our simple feature engineering model, *Bag of Features for Link Prediction* (BFLP). Our primary aim in this approach is to furnish our ML classifier with a comprehensive set of relevant and potentially valuable information concerning node pairs. We intentionally opt to allow the classifier the discretion to determine which features to leverage, depending on the dataset at hand. While existing literature typically employs these features individually or in pairs, our intuition leads us to believe that by aggregating all of this information, the machine learning classifier can make finer determinations within the feature space. Furthermore, these features can synergistically reinforce each other, resulting in a more robust and accurate model.

In this context, our collection of features can be categorized into two distinct types: *Structural Features* and *Domain Features*. Structural features draw upon the inherent graph structure and neighborhood information, while domain features leverage similarity measures derived from the node features provided.

STRUCTURAL FEATURES

Within our *Bag of Features* model, we incorporate a range of structural features. While a subset of these features corresponds to established similarity indices as detailed in Section 2.2, along with their generalizations, there are also novel features we introduce. These newly defined features are designed to capture finer insights from the local neighborhoods of node pairs.

The established features are Jacard Index, Salton Index, Sorensen Index, and Adamic Adar (Section 2.2). Furthermore, for Jacard, Salton, and Sorensen indices, we use their natural generalizations for 3-neighborhood versions as well as new features like length-k paths and distance index.

*Length-k paths index:* For a given $u, v \in \mathcal{V}$, we define *length-k paths index* $\mathcal{L}_k(u, v)$ as the total number of length-$k$ paths between the nodes $u$ and $v$. Notice that length-2 paths index is the same with common neighbors index, i.e. $\mathcal{L}_2(u, v) = \mathcal{CM}(u, v) = |\mathcal{N}(u) \cap \mathcal{N}(v)|$.

*3-Jaccard Coefficient* is a slight modification of the original Jaccard Coefficient by using $\mathcal{L}_3(u, v)$ the length 3-paths (squares) instead of $\mathcal{L}_2(u, v)$ length-2 paths (triangles) in our definition.

$$\mathcal{J}^3(u, v) = \frac{\mathcal{L}_3(u, v)}{|\mathcal{N}(u) \cup \mathcal{N}(v)|}$$

By using a similar idea, we implement a comparable adaptation to both the Salton and the Sorensen Index, resulting in the following formulation:

*3-Salton Index:*
$$\mathcal{S}_a^3(u, v) = \frac{\mathcal{L}_3(u, v)}{\sqrt{k_u k_v}}$$

*3-Sorensen Index:*
$$\mathcal{S}_o^3(u, v) = \frac{2\mathcal{L}_3(u, v)}{k_u + k_v}$$

*Distance index:* For a given $u, v \in \mathcal{V}$, we define *distance index* $\mathcal{D}(u, v)$ as the length of the shortest path between $u$ and $v$ in $\mathcal{G}$. Note that when computing $D(u, v)$, we remove the edge between $u$ and $v$ from the graph if $u$ and $v$ are adjacent nodes. Therefore, $\mathcal{D}(u, v) \geq 2$ for any $u \neq v \in \mathcal{V}$. The main motivation to define this index in this particular way is that in the test set, a priori, there won't be an edge between the nodes. Therefore, during training ML classifier, this distance index provides valuable information to the ML classifier to distinguish positive and negative edges when combined with other features.

Hence, for a given node pair $u, v \in \mathcal{V}$, we produce ten structural features as follows $\mathcal{J}(u, v), \mathcal{S}_a(u, v), \mathcal{S}_o(u, v), \mathcal{J}^3(u, v), \mathcal{S}_a^3(u, v), \mathcal{S}_o^3(u, v), \mathcal{AA}(u, v), \mathcal{L}_2(u, v), \mathcal{L}_3(u, v)$ and $\mathcal{D}(u, v)$.

DOMAIN FEATURES

Next, we describe our domain features. Contrary to our structural features, our domain features do not use graph structure, but only the feature vectors. For a given graph with node features $\mathcal{G} = (\mathcal{V}, \mathcal{E}, \mathcal{X})$, let $\mathbf{X}_u \in \mathbb{R}^m$ represent the feature vector for the node $u \in \mathcal{V}$. In the following, for a given node pair $u, v \in \mathcal{V}$, we extract our domain features $\alpha(u, v)$ by using the similarity/dissimilarity of these node feature vectors $\mathbf{X}_u$ and $\mathbf{X}_v$.

We have several forms of domain features depending on the format of the feature vector. We group them into three categories.

$\diamond$ **$\mathbf{X}_u$ is binary vector: $\mathbf{X}_u^i \in \{0, 1\}$**

In this case, we naturally interpret this as every binary digit in the vector $\mathbf{X}_u$ represents the existence or nonexistence of a property. For example, if $\mathcal{G}$ represents a citation network, where nodes represent papers, $\mathbf{X}_u$ can be a binary vector representing the existence or nonexistence of previously chosen keywords in the paper $u$. We define two similarity measures between $\mathbf{X}_u$ and $\mathbf{X}_v$.

*Common Digits:* If $\mathbf{X}_u$ is a binary vector, we define our *Common Digits* domain feature $\mathfrak{CD}(u, v)$ as the number of matching "1"s in the vectors $\mathbf{X}_u$ and $\mathbf{X}_v$. i.e.,

$$\mathfrak{CD}(u, v) = \#\{i \mid \mathbf{X}_u^i = \mathbf{X}_v^i = 1\}$$

Note that one can similarly define an analogous feature vector as the number of common "0"s to emphasize the common absent properties.

*Normalized Common Digits:* This domain feature is a slight variation of the previous one with some normalization factor. In particular, if $\mathbf{X}_u$ and $\mathbf{X}_v$ have only a few positive digits in their vectors, having an equal number of common digits would result in them being considered more similar, in contrast to node pairs with numerous positive digits. We normalize this feature by dividing it by the total number of positive digits in both vectors $\mathbf{X}_u$ and $\mathbf{X}_v$ (not counting the common positive digits twice).

$$\widehat{\mathfrak{CD}}(u, v) = \frac{\#\{i \mid \mathbf{X}_u^i = \mathbf{X}_v^i = 1\}}{\#\{j \mid (\mathbf{X}_u + \mathbf{X}_v)^j \geq 1\}}$$

Notice that the vector $\mathbf{X}_u + \mathbf{X}_v$ would have only $0, 1$, and $2$ digits where $2$s represent the common positive digits in $\mathbf{X}_u$ and $\mathbf{X}_v$. i.e., $\mathfrak{CD}(u, v) = \#\{j \mid (\mathbf{X}_u + \mathbf{X}_v)^j = 2\}$

$\diamond$ **$\mathbf{X}_u^i$ takes finitely many values: $\mathbf{X}_u^i \in \{1, 2, \ldots, m\}$**

In this case, we make the assumption that a particular entry denoted as $\mathbf{X}_u^i$ can assume a finite set of distinct values, such as $\mathbf{X}_u^i \in 1, 2, \ldots, m$. In such cases, we interpret this specific entry as representing some sort of "class information" pertaining to a particular node feature of $u$. Notably, if a node classification is provided in the data, we take this information into account within this context. For instance, in citation networks, this data could correspond to the academic field of the paper (e.g., Mathematics, Computer Science, History) as a feature of the node. Within this category, we establish two distinct domain features.

*Common Class:* This is a simple binary domain feature to detect if the "classes" are the same or not. In particular, the *Common Class* feature is $1$ if they are the same, $0$ otherwise, i.e. $\mathfrak{CC}(u, v) = 1$ if $\mathbf{X}_u^i = \mathbf{X}_v^i$, and $\mathfrak{CC}(u, v) = 0$ if $\mathbf{X}_u^i \neq \mathbf{X}_v^i$. Our intuition here is that being in the same "class" for two nodes is an important factor in deciding if there is a link between them.

*Class Identifier:* This time we aim to give the raw "class" information to our ML classifier as the interaction between different classes might be an important factor for link prediction. In other words, during training ML classifier might detect the pattern between certain "classes" given other features which can be key information for link prediction, i.e. $\mathfrak{CI}(u, v) = (\mathbf{X}_u^i, \mathbf{X}_v^i)$ where $\mathbf{X}_u^i \in \{1, 2, \ldots, m\}$ represents the class of $u$, e.g., $\mathfrak{CI}(u, v) = (3, 5)$ where $3$ and $5$ represents the class labels for $u$ and $v$, respectively.

$\diamond$ **$\mathbf{X}_u$ is a real-valued vector: $\mathbf{X}_u \in \mathbb{R}^m$**

When $\mathbf{X}_u$ is represented as a real-valued vector, it inherently serves as a node embedding within a feature space $\mathbb{R}^m$. Therefore, the similarity or dissimilarity between two nodes is intuitively

associated with the separation between these embeddings. We employ two distinct types of distance measurements as domain features.

$L^1$ *Distance:* Simply, we use $L^1$-norm (Manhattan metric) in the feature space $\mathbb{R}^m$. If $\mathbf{X}_u = [a_1 \ a_2 \ \ldots \ a_m]$ and $\mathbf{X}_v = [b_1 \ b_2 \ \ldots \ b_m]$, we define

$$\mathfrak{D}(u,v) = \mathbf{d}(\mathbf{X}_u, \mathbf{X}_v) = \sum_{i=1}^{m} |a_i - b_i|$$

*Cosine Distance:* Another popular distance formula using some normalization is the cosine distance/similarity. We define our cosine distance feature as

$$\mathfrak{D}^{\mathfrak{c}}(u,v) = \frac{\mathbf{X}_u \cdot \mathbf{X}_v}{\|\mathbf{X}_u\|.\|\mathbf{X}_v\|}$$

We would like to note that numerous node attributes may result from a fusion of these three categories. In such instances, we incorporate all of them by dissecting the node feature vector based on their respective subtypes, subsequently acquiring the corresponding domain features for the node pairs.

## 4 EXPERIMENTS

### 4.1 EXPERIMENTAL SETUP

**Datasets.** In our experiments, we used six benchmark datasets for link prediction tasks. All the datasets are used in the transductive setting like most other baselines in the domain. The dataset statistics are given in Table 1.

The citation network datasets, namely CORA, CITESEER and PUBMED are introduced in (Yang et al., 2016), and they serve as valuable benchmark datasets for research in the field of semi-supervised learning with graph representation learning. Within these datasets, individual nodes correspond to distinct documents, while the edges between them symbolize citation links, elucidating the interconnectedness of scholarly works within these domains.

In the context of co-purchasing networks, the benchmark datasets, PHOTO, and COMPUTERS, are introduced in (Shchur et al., 2018) representing the sales network at Amazon. In these networks, nodes correspond to various products, while edges signify the frequent co-purchasing of two products. The primary objective of this study is to leverage product reviews, represented as bag-of-words node features, to establish a mapping between individual goods and their respective product categories, thus addressing a fundamental categorization task within the context of these interconnected networks.

Finally, the OGBL-COLLAB dataset is a part of the library of large benchmark datasets, namely Open Graph Benchmark (OGB) collection (Hu et al., 2020; 2021a). This is an undirected graph, representing a subset of the collaboration network between authors indexed by Microsoft Academic Graph (MAG) (Wang et al., 2020). Each node represents an author and edges indicate the collaboration between authors. All nodes come with 128-dimensional features, obtained by averaging the word embeddings of papers that are published by the authors. All edges are associated with two meta-information: the year and the edge weight, representing the number of co-authored papers published in that year. The graph can be viewed as a dynamic multi-graph since there can be multiple edges between two nodes if they collaborate in more than one year.

**Experiment Settings.** In the experimental methodology used for analyzing the datasets CORA, CITESEER, and PUBMED, we followed one of the common settings used for these datasets (Zhao et al., 2022). Specifically, 70% of the data is allocated for positive training purposes, while 10% is set aside for validation and another 20% for testing. Correspondingly, an equal number of non-existing pairs (negative set) are randomly selected for each of these sets.

To ensure accurate training results, graph structures in both the positive validation and test sets are masked before training on the positive training dataset takes place. This process is repeated a total of 10 times. On the other hand, for datasets COMPUTERS and PHOTO, we use a different split configuration with ratios of 85/5/10 respectively following (Guo et al., 2022); and we repeated

Table 1: Characteristics of our benchmark datasets for link prediction. FV Type represents the type of the node feature vector provided.

| Datasets | Nodes | Edges | Classes | Features | FV Type |
|---|---|---|---|---|---|
| CORA | 2,708 | 5,429 | 7 | 1,433 | Binary |
| CITESEER | 3,312 | 4,732 | 6 | 3,703 | Binary |
| PUBMED | 19,717 | 44,338 | 3 | 500 | Binary |
| PHOTO | 7,650 | 119,081 | 8 | 745 | Binary |
| COMPUTERS | 13,752 | 245,861 | 10 | 767 | Binary |
| OGBL-COLLAB | 235,868 | 1,285,465 | – | 128 | Real |

Table 2: Total number of features used for each dataset in our model.

| | CORA | CITESEER | PUBMED | PHOTOS | COMPUTERS | OGBL-COLLAB |
|---|---|---|---|---|---|---|
| # Features | 15 | 15 | 15 | 15 | 15 | 28 |

this specific process five times. It should be noted that regarding the OGBL-COLLAB dataset, the sets including both positive (training/validation/test) as well as negative (validation/test) are predefined within the dataset. The negative training set is randomly selected through a repetitive process conducted 10 times.

**Feature Sets.**    In all datasets except OGBL-COLLAB, we used the same feature sets described in Section 3.2. Since OGBL-COLLAB is dynamic and weighted, we needed to adjust our features in this dataset to adapt our method to this context. The total number of features used for each dataset is given in Table 2. We gave the details of our feature sets for each dataset in Appendix A.

**Metrics.**    There are various performance metrics used in the domain. While the most popular one is AUC (The area under the receiver operating characteristic curve), depending on the context, other performance metrics are also used. The second metric we use is AP (the area under the precision-recall rate curve). Our final metric is Hits@K. In particular, after ranking each true link among a set of 100,000 randomly sampled negative links, we count the ratio of positive links that are ranked at K-place or above (Hits@K) (Hu et al., 2021b). For OGB datasets, the performance metrics are suggested by the creators of the datasets. For OGB-COLLAB, we use the predefined metric for this dataset, i.e., Hits@50.

**Hyperparameter Settings.**    In our study, we utilize XGBoost as our machine learning tool. We establish the objective function as rank:pairwise with logloss as the evaluation metric. When evaluating results using the AUC metric, we configure key parameters: maximum tree depth 3, learning rate 0.1, subsample ratio 0.8, colsample bytree ratio 0.8, the number of estimators 500, and the regularization parameter lambda 5.0. Nonetheless, for the AP metric, we set the maximum tree depth to 7 while keeping other hyperparameters unchanged. Conversely, for the challenging metric Hits@20, we adjust the maximum tree depth to 5, the subsample ratio to 0.5, and the number of estimators to 200 and maintain the remaining parameters unchanged. For metric Hits@50 we reset maximum tree depth to 11, learning rate 0.5, and lambda to 1.0, while the othe hyperparameters are kept same as the hyperparameters utilized for AUC metric.

**Implementation and Runtime.**    We ran experiments on a single machine with 12th Generation Intel Core i7-1270P vPro Processor (E-cores up to 3.50 GHz, P-cores up to 4.80 GHz), and 32Gb of RAM (LPDDR5-6400MHz). While end-to-end runtime (computing feature vectors and ML classifier) for OGBL-COLLAB is 30 minutes, the most time-consuming dataset is COMPUTERS, requiring 18 hours of parallel task computations. The computational complexities of our similarity indices are $\mathcal{O}(|\mathcal{V}|.k^3)$ where $|\mathcal{V}|$ is the total number of nodes and $k$ is the maximum degree in the network (Martínez et al., 2016). We provide our code at the link[1]

---

[1]Code link: `https://github.com/workrep20232/LinkPrediction`

Table 3: Link prediction performances measured by AUC and Hits@20. Best performance and second best performance are marked with **bold blue** and blue, respectively.

| Models | CORA | | CITESEER | | PUBMED | |
|---|---|---|---|---|---|---|
| | AUC | Hits@20 | AUC | Hits@20 | AUC | Hits@20 |
| Node2Vec | $84.49 \pm_{0.49}$ | $49.96 \pm_{2.51}$ | $80.00 \pm_{0.68}$ | $47.78 \pm_{1.72}$ | $80.32 \pm_{0.29}$ | $39.19 \pm_{1.02}$ |
| MVGRL | $75.07 \pm_{3.63}$ | $19.53 \pm_{2.64}$ | $61.20 \pm_{0.55}$ | $14.07 \pm_{0.79}$ | $80.78 \pm_{1.28}$ | $14.19 \pm_{0.85}$ |
| VGAE | $88.68 \pm_{0.40}$ | $45.91 \pm_{3.38}$ | $85.35 \pm_{0.60}$ | $44.04 \pm_{4.86}$ | $95.80 \pm_{0.13}$ | $23.73 \pm_{1.61}$ |
| SEAL | $92.55 \pm_{0.50}$ | $51.35 \pm_{2.26}$ | $85.82 \pm_{0.44}$ | $40.90 \pm_{3.68}$ | $96.36 \pm_{0.28}$ | $28.45 \pm_{3.81}$ |
| GCN | $90.25 \pm_{0.53}$ | $49.06 \pm_{1.72}$ | $71.47 \pm_{1.40}$ | $55.56 \pm_{1.32}$ | $96.33 \pm_{0.80}$ | $21.84 \pm_{3.87}$ |
| GSAGE | $90.24 \pm_{0.34}$ | $53.54 \pm_{2.96}$ | $87.38 \pm_{1.39}$ | $53.67 \pm_{2.94}$ | $96.78 \pm_{0.11}$ | $39.13 \pm_{4.41}$ |
| JKNet | $89.05 \pm_{0.67}$ | $48.21 \pm_{3.86}$ | $88.58 \pm_{1.78}$ | $55.60 \pm_{2.17}$ | $96.58 \pm_{0.23}$ | $25.64 \pm_{4.11}$ |
| CFLP | $93.05 \pm_{0.24}$ | **$65.57 \pm_{1.05}$** | $92.12 \pm_{0.47}$ | $68.09 \pm_{1.49}$ | **$97.53 \pm_{0.17}$** | **$44.90 \pm_{2.00}$** |
| BFLP | **$94.54 \pm_{0.30}$** | $61.21 \pm_{2.63}$ | **$95.01 \pm_{0.35}$** | **$72.03 \pm_{1.47}$** | $94.73 \pm_{0.15}$ | $42.12 \pm_{2.00}$ |

Table 4: Link prediction performances measured by AUC and AP. Best performance and second best performance are marked with **bold blue** and blue, respectively.

| Models | PHOTO | | COMPUTERS | |
|---|---|---|---|---|
| | AUC | AP | AUC | AP |
| GAE | $96.5 \pm_{0.03}$ | $96.2 \pm_{0.02}$ | $92.5 \pm_{0.03}$ | $92.8 \pm_{0.02}$ |
| VGAE | $95.2 \pm_{0.04}$ | $94.9 \pm_{0.04}$ | $92.5 \pm_{0.04}$ | $92.8 \pm_{0.05}$ |
| ARGA | $94.3 \pm_{0.02}$ | $93.7 \pm_{0.02}$ | $94.2 \pm_{0.02}$ | $94.3 \pm_{0.01}$ |
| ARVGA | $93.7 \pm_{0.04}$ | $92.5 \pm_{0.05}$ | $93.7 \pm_{0.01}$ | $93.1 \pm_{0.01}$ |
| DBGAN | $96.3 \pm_{0.01}$ | $95.8 \pm_{0.01}$ | $94.6 \pm_{0.01}$ | $94.2 \pm_{0.02}$ |
| MSVGAE | $96.7 \pm_{0.01}$ | $96.3 \pm_{0.01}$ | $95.1 \pm_{0.02}$ | $94.6 \pm_{0.01}$ |
| BFLP | **$99.0 \pm_{0.02}$** | **$98.9 \pm_{0.02}$** | **$98.6 \pm_{0.04}$** | **$98.7 \pm_{0.05}$** |

Table 5: Hits@50 Performances for OGBL-COLLAB

| N2Vec | MF | MLP | GCN | GAT | GSAGE | SEAL | BUDDY | Neo-GNN | NCN | NCNC | NBFNet | OGB Leader | Ours |
|---|---|---|---|---|---|---|---|---|---|---|---|---|---|
| 49.06 | 41.81 | 35.81 | 54.96 | 55.00 | 59.44 | 63.37 | 64.59 | 66.13 | 63.86 | 65.97 | OOM | 70.96 | **76.50** |

## 4.2 RESULTS AND DISCUSSION

**Baselines.** We compare the link prediction performance of our model BFLP against the common embedding methods and GNNs. The embedding methods include Matrix Factorization (MF) (Menon & Elkan, 2011), MLP and Node2Vec (Grover & Leskovec, 2016), which are used to learn low-dimensional node embeddings to predict the likelihood of node pairs existing. For GNNs, we include GCN (Kipf & Welling, 2016a), GraphSAGE (Hamilton et al., 2017), GAT (Veličković et al., 2018), GAE and VGAE (Kipf & Welling, 2016b), SEAL (Zhang & Chen, 2018), JKNet (Xu et al., 2018), ARGA and ARVGA (Pan et al., 2018), MVGRL (Hassani & Khasahmadi, 2020), DBGAN (Zheng et al., 2020), LGLP (Cai et al., 2021), MSVGAE (Guo et al., 2022), CFLP (Zhao et al., 2022). For more details on these baselines, see (Li et al., 2023). For OGB-COLLAB, we used the baseline performances from (Li et al., 2023) and further reported the performance of the current leader (Wang et al., 2022) at OGB Leaderboard [2] as of September 28, 2023.

**Results.** We give our results in the Tables 3 to 5. Further results can be found in the appendix (Tables 8 to 10). It's clear that our *Bag of Features* model outperforms most of the current benchmarks across the six datasets, with the exception of PUBMED. In the PUBMED dataset, our model closely trails the top performer, securing the second position for the Hits@20 metric. Impressively, in datasets like CITESEER, PHOTO, and COMPUTERS, our computationally efficient model leads in every metric. What's particularly noteworthy is its performance on the widely recognized benchmark

---

[2] https://ogb.stanford.edu

for GNN models, the OGB-COLLAB dataset. Despite OGB being a benchmark for the best GNN models, our straightforward model surpasses its competition, marking a remarkable achievement.

**Ablation Study.** In our ablation study (Table 6), we evaluated the relative contributions of our structural and domain features. The outcomes are varied: domain features demonstrate greater significance in datasets like CORA and CITESEER, whereas structural features take precedence in datasets such as PHOTO and COMPUTERS. However, a consistent observation across all datasets is the synergistic effect of these features. When combined, they invariably enhance the overall performance. We reported further ablation studies in the appendix for the importance of individual features (Table 7) and the performance of different ML classifiers (Table 11).

Table 6: **Ablation Study.** AUC results for our model for different feature subsets.

| Features | CORA | CITESEER | PUBMED | PHOTO | COMPUTERS |
|---|---|---|---|---|---|
| Structural only | 83.98 $\pm0.53$ | 75.96 $\pm0.96$ | 87.94 $\pm0.29$ | 98.7 $\pm0.03$ | 98.2 $\pm0.04$ |
| Domain only | 90.87 $\pm0.28$ | 92.21 $\pm0.42$ | 87.56 $\pm0.26$ | 93.1 $\pm0.15$ | 89.2 $\pm0.05$ |
| All Features | **94.54**$\pm$ **0.30** | **95.01**$\pm$ **0.35** | **94.73**$\pm$ **0.15** | **99.0**$\pm$ **0.02** | **98.6**$\pm$ **0.04** |

**Discussion.** Our experiments show that our simple ML model, which is a combination of traditional feature engineering methods with a tree-based ML classifier, outperforms or gives on-par performance with most of the current GNN models in benchmark datasets. We would like to note that another feature engineering method (Adamic-Adar & Edge Proposal Set) by Singh et al. (2021) has a very high ranking in the OGBL-COLLAB leaderboard. These findings may be seen as unexpected since GNN models are generally perceived as the frontrunners in graph representation learning. While deep learning models have established dominance in areas like computer vision and NLP, recent theoretical studies suggest that the same might not hold true for GNNs. Our data underscores that there is a pressing need for fresh and innovative approaches within the realm of GNNs to tackle unique challenges in graph representation learning and enhance performance.

**Limitations.** The main limitation in our approach comes from the customization of the domain features depending on the format of the node feature vectors and the context. Unfortunately, there is no general rule in this part, as the context of the dataset plays a crucial part in extracting useful domain features. However, considering node features as node embedding in the feature space, it might be possible to use GNNs to obtain the most effective domain feature vectors by formulating the question in terms of learnable parameters. In our future projects, we aim to further explore this direction.

## 5    CONCLUSION

In this paper, motivated by the theoretical studies questioning the expressivity of GNNs, we have compared the performance of state-of-the-art GNN models with standard feature engineering methods in link prediction tasks. Our computationally efficient model, which builds on the established feature extraction methods in the domain, outperforms or gives highly competitive results with state-of-the-art GNNs in both small/large benchmark datasets. Our results empirically indicate that GNNs in their current form are far from their counterparts in computer vision and natural language processing. In our forthcoming projects, we aim to integrate our feature engineering approach into GNNs to direct them to learn more robust feature representations, thereby enhancing their overall performance.

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

# A FEATURE SETS FOR EACH DATASET

## FEATURES FOR ALL DATASETS EXCEPT OGBL-COLLAB

In CORA, CITESEER, PUBMED, PHOTO, and COMPUTERS, we used the same feature set as follows:

**Structural Features:** $\mathcal{J}(u,v), \mathcal{S}_a(u,v), \mathcal{S}_o(u,v), \mathcal{J}^3(u,v), \mathcal{S}_a^3(u,v), \mathcal{S}_o^3(u,v), \mathcal{AA}(u,v), \mathcal{L}_2(u,v), \mathcal{L}_3(u,v)$ and $\mathcal{D}(u,v)$

**Domain Features:** $\mathfrak{CC}(u,v), \mathfrak{CI}(u,v), \mathfrak{CD}(u,v), \widehat{\mathfrak{CD}}(u,v)$

In particular, in our *Bag of Features*, we have 10 structural features and 5 domain features ($\mathfrak{CI}(u,v)$ is 2-dimensional). The details of these features are given in Section 2.2. The importance of each feature for each datasets is given in Table 7.

## FEATURES FOR OGBL-COLLAB

The OGBL-COLLAB dataset is a time-varying dataset, spanning between years 1963 to the year 2019, where the positive training set spans between years 1963 and 2017, the positive validation set is set the node pairs appearing in the year 2018, and year 2019 is the positive test set. Each link has a weight which is the number of collaborations that occur between the authors pair for the given year. Since it is dynamic and weighted, we needed to adjust our features in this dataset to adapt our method to this context.

Let $w^y(u,v)$ be the number of collaborations that occur between nodes $u$ and $v$ in the year $y$. For simplicity let us define the total collaborations of a pair $(u,v)$ through all years as

$$\mathcal{W}(u,v) = \{\sum w^y(u,v) : 1963 \leq y \leq 2017\}.$$

We define the number of collaborations of the pair $(u,v)$ between the years 2007 and 2017 as

$$\mathcal{W}_{10}(u,v) = \{\sum w^y(u,v) : 2007 \leq y \leq 2017\},$$

and between years 2012 and 2017 as

$$\mathcal{W}_5(u,v) = \{\sum w^y(u,v) : 2012 \leq y \leq 2017\}.$$

Finally, considering $\mathcal{G} = \{(u,v) : \text{the collaboration between } u \text{ and } v \text{ occurs in years 2007 to 2017}\}$,

$$\mathcal{A}(u) = \{\sum \mathcal{W}_{10}(u,x) : x \in \Gamma(u)\}.$$

**Author's Oldest Paper Index:** For this feature we track down the year of the earliest paper of each author. If this year is before year 1985, we assign the value 0, and 1 otherwise.

**Author's Newest Paper Index:** For this feature we track down the year of the latest paper of each author. If this year is before year 1985, we assign the value 0, and 1 otherwise.

**All Time Collaborations:** For each pair (u,v) we add-up the weights of the pair through each year for which the link exists, which is $\mathcal{W}(u,v)$.

**10-Year Collaborations:** For each pair (u,v) we add-up the weights of the pair through each year, from 2007 to 2017, for which the link exists, $\mathcal{W}_{10}(u,v)$.

**5-Year Collaborations:** For each pair (u,v) we add-up the weights of the pair through each year, from 2012 to 2017, for which the link exists $\mathcal{W}_5(u,v)$.

**All Time Common Collaborators:** For each pair (u,v) we find the neighborhood of node u and node v over the graph created by combining all the years between 1963 and 2017, and the we take the intersection of the neighborhoods.

**10-Year Common Collaborators:** For each pair (u,v) we find the neighborhood of node u and node v over the graph created by combining the years between 2007 and 2017, and the we take the intersection of the neighborhoods.

**5-Year Common Collaborators:** For each pair (u,v) we find the neighborhood of node u and node v over the graph created by combining the years between 2012 and 2017, and the we take the intersection of the neighborhoods.

**Preferential Attachment:** We evaluate Preferential Attachment (Barabâsi et al., 2002) over the graph created by combining the years between 2007 and 2017, which formula is

$$PA(u, v) = \mathcal{A}(u) \cdot \mathcal{A}(v).$$

**w-Adamic Andar:** Considering the graph $\mathcal{G}$,

$$\mathcal{A}\mathcal{A}^w(u, v) = \sum_{z \in \mathcal{N}(u) \cap \mathcal{N}(v)} \frac{1}{\log |\mathcal{A}(z)|}.$$

**w-Jaccard Index:** Considering the graph $\mathcal{G}$,

$$\mathcal{J}^w(u, v) = \frac{\{\sum \mathcal{W}_{10}(u, z) + \mathcal{W}_{10}(z, v) : z \in \mathcal{N}(u) \cap \mathcal{N}(v)\}}{\{\sum \mathcal{W}_{10}(u, x) + \mathcal{W}_{10}(x, v) : x \in \mathcal{N}(u) \cup \mathcal{N}(v)\}}.$$

**w-Salton Index:** Considering the graph $\mathcal{G}$,

$$\mathcal{S}_a^w(u, v) = \frac{\{\sum \mathcal{W}_{10}(u, z) + \mathcal{W}_{10}(z, v) : z \in \mathcal{N}(u) \cap \mathcal{N}(v)\}}{\sqrt{\mathcal{A}(u)\mathcal{A}(v)}}.$$

**Shortest Path Length:** $\mathcal{D}(u, v)$ over the graph $\mathcal{G}$.

For each node, a 128 dimensional feature vector of word embedings is provided. We set up three features to utilize them. These features are:

**Common Embedding:** For given word embedding $\mathbf{X}$, we define our *Common Embedding* domain feature $\mathfrak{C}\mathfrak{E}(u, v)$ as the number of matching "1"s in the vectors $\mathbf{X}_u$ and $\mathbf{X}_v$. i.e.,

$$\mathfrak{C}\mathfrak{E}(u, v) = \#\{i \mid \mathbf{X}_u^i = \mathbf{X}_v^i\}$$

$L^1$ **Distance:** Simply, we use $L^1$-norm (Manhattan metric) in the feature space $\mathbb{R}^m$. If $\mathbf{X}_u = [a_1 \ a_2 \ \ldots \ a_m]$ and $\mathbf{X}_v = [b_1 \ d_2 \ \ldots \ b_m]$, we define $\mathfrak{D}(u, v) = \mathbf{d}(\mathbf{X}_u, \mathbf{X}_v) = \sum_{i=1}^m |a_i - b_i|$.

**Cosine Distance:** Another popular distance formula using some normalization is the cosine distance/similarity. We define our cosine distance feature as $\mathfrak{D}^{\mathfrak{c}}(u, v) = \dfrac{\mathbf{X}_u \cdot \mathbf{X}_v}{\|\mathbf{X}_u\| . \|\mathbf{X}_v\|}$

**Year-wise label:** For node pair $(u, v)$, we define $\mathcal{L}\mathcal{A}_y = 1$ if the pair exists in year $y$, and 0 otherwise. In our experiments we iterate $y$ between years 2007 to 2016.

## B  FURTHER EXPERIMENTS

### B.1  FEATURE IMPORTANCE

In Table 7, we present the importance of individual features in our model across various datasets. Our feature set proves to be highly versatile, effectively adapting to the unique characteristics of each dataset. Notably, certain features exhibit substantial importance in specific datasets, while their impact is minimal in others. Additionally, the synergy between domain and structural features emerges as a key factor contributing to enhanced performance.

Table 7: **Feature Importance.** For each dataset, the importance weights of our features for our ML classifier XGBoost.

| CORA | | CITESEER | | PUBMED | | COMPUTERS | | PHOTO | |
|---|---|---|---|---|---|---|---|---|---|
| Same Class | 0.4749 | # 3-paths | 0.2630 | # 3-paths | 0.4179 | A. Adar | 0.4974 | A. Adar | 0.4708 |
| # 3-paths | 0.1255 | Same Class | 0.1582 | Same Class | 0.1574 | # 2-paths | 0.0963 | Salton | 0.1373 |
| # 2-paths | 0.1199 | # Com. Digits | 0.1237 | 3-Jaccard | 0.1427 | Salton | 0.0956 | 3-Salton | 0.1319 |
| 3-Jaccard | 0.0601 | # 2-paths | 0.1176 | distance | 0.0966 | 3-Salton | 0.0926 | # 2-paths | 0.0651 |
| A. Adar | 0.0538 | N. Com. Digits | 0.0992 | 3-Salton | 0.0604 | Jaccard | 0.0788 | Same Class | 0.0518 |
| # Com. Digits | 0.0355 | 3-Jaccard | 0.0817 | N. Com. Digits | 0.0487 | Same Class | 0.0344 | # 3-paths | 0.0398 |
| N. Com. Digits | 0.0346 | A. Adar | 0.0437 | # 2-paths | 0.0202 | # 3-paths | 0.0299 | distance | 0.0366 |
| distance | 0.0308 | 3-Salton | 0.0233 | A. Adar | 0.0151 | distance | 0.0278 | 3-Sorensen | 0.0210 |
| 3-Salton | 0.0117 | Jaccard | 0.0225 | # Com. Digits | 0.0104 | Class(u) | 0.0125 | Class(v) | 0.0131 |
| Jaccard | 0.0110 | distance | 0.0213 | 3-Sorensen | 0.0094 | # Com. Digits | 0.0106 | Class(u) | 0.0121 |
| Class(u) | 0.0108 | 3-Sorensen | 0.0107 | Class(v) | 0.0073 | Class(v) | 0.0079 | # Com. Digits | 0.0074 |
| 3-Sorensen | 0.0087 | Class(u) | 0.0106 | Class(u) | 0.0062 | N. Com. Digits | 0.0044 | 3-Jaccard | 0.0048 |
| Class(v) | 0.0085 | Class(v) | 0.0105 | Jaccard | 0.0031 | 3-Jaccard | 0.0044 | Jaccard | 0.0032 |
| Salton | 0.0079 | Salton | 0.0080 | Salton | 0.0029 | 3-Sorensen | 0.0039 | Sorensen | 0.0030 |
| Sorensen | 0.0063 | Sorensen | 0.0061 | Sorensen | 0.0019 | Sorensen | 0.0036 | N. Com. Digits | 0.0021 |

## B.2 NEW BASELINES

We add additional SOTA GNN model performances for our benchmark datasets, i.e. BUDDY (Chamberlain et al., 2022), Neo-GNN (Yun et al., 2021), NCN and NCNC (Wang et al., 2023), and NBFNet (Zhu et al., 2021). Note that the accuracy results are taken from different references (Li et al., 2023; Guo et al., 2022) with different splits.

Table 8: AUC performances of additional GNN models with different splits from (Li et al., 2023).

| Model | Split | CORA | CITESEER | PUBMED |
|---|---|---|---|---|
| BUDDY | 85:05:10 | 95.06±0.36 | 96.72±0.26 | 98.20±0.05 |
| Neo-GNN | 85:05:10 | 93.73±0.36 | 94.89±0.60 | 98.71±0.05 |
| NCN | 85:05:10 | 96.76±0.18 | 97.04±0.26 | 98.98±0.04 |
| NCNC | 85:05:10 | 96.90±0.28 | 97.65±0.30 | 99.14±0.03 |
| NBFNet | 85:05:10 | 92.85±0.17 | 91.06±0.15 | 98.34±0.02 |
| **Ours** | 70:10:20 | 94.54±0.30 | 95.01±0.35 | 94.73±0.15 |

Table 9: AUC performances of additional GNN models with different splits from (Guo et al., 2022).

| Methods | Split | Cora | Citeseer |
|---|---|---|---|
| GAE | 85:05:10 | 91.5±0.02 | 91.0±0.04 |
| VGAE | 85:05:10 | 91.5±0.04 | 91.2±0.03 |
| ARGA | 85:05:10 | 91.5±0.03 | 92.8±0.03 |
| ARVGA | 85:05:10 | 93.1±0.03 | 92.5±0.01 |
| DBGAN | 85:05:10 | 94.5±0.01 | 94.5±0.04 |
| MSVGAE | 85:05:10 | 95.3±0.05 | 95.4±0.03 |
| Ours | 70:10:20 | 94.5±0.30 | 95.0±0.35 |

Table 10: Hits@50 performances with SD for OGBL-COLLAB Dataset.

| Model | Hits@50 |
|---|---|
| Node2Vec | $49.06 \pm 1.04$ |
| MF | $41.81 \pm 1.67$ |
| MLP | $35.81 \pm 1.08$ |
| GCN | $54.96 \pm 3.18$ |
| GAT | $55.00 \pm 3.28$ |
| GSAGE | $59.44 \pm 1.37$ |
| SEAL | $63.37 \pm 0.69$ |
| BUDDY | $64.59 \pm 0.46$ |
| Neo-GNN | $66.13 \pm 0.61$ |
| NCN | $63.86 \pm 0.51$ |
| NCNC | $65.97 \pm 1.03$ |
| NBFNet | OOM |
| OGB Leader | $70.96 \pm 0.55$ |
| Ours | $\mathbf{76.50 \pm 0.27}$ |

## B.3 PERFORMANCE OF ML CLASSIFIERS

In Table 11, we present the performance of various ML classifiers using our feature vectors. The consistent performance across different classifiers indicates that our features are model-agnostic, and provide robust information regarding the likelihood of establishing a link between node pairs.

Table 11: **ML Classifiers.** AUC results for our model for different ML Classifiers.

| ML Classifier | CORA | CITESEER | PUBMED | PHOTO | COMPUTERS |
|---|---|---|---|---|---|
| Logistic Regression | $93.58 \pm 0.39$ | $94.45 \pm 0.36$ | $93.83 \pm 0.15$ | $98.68 \pm 0.04$ | $98.17 \pm 0.03$ |
| Naive Bayes | $93.32 \pm 0.37$ | $93.24 \pm 0.58$ | $92.93 \pm 0.18$ | $97.14 \pm 0.08$ | $96.49 \pm 0.09$ |
| XGBoost | $\mathbf{94.54 \pm 0.30}$ | $\mathbf{95.01 \pm 0.35}$ | $\mathbf{94.73 \pm 0.15}$ | $\mathbf{98.99 \pm 0.02}$ | $\mathbf{98.65 \pm 0.04}$ |

## B.4 HETEROPHILIC DATASETS

We report the performance of our model in heterophilic datasets in Table 12. We'd like to highlight that our model isn't tailored specifically for heterophilic datasets; rather, it's designed to effectively handle both homophilic and heterophilic datasets. While our model demonstrates superior performance compared to standard GNN models, it might not match the performance of GNN models specifically optimized for this particular setting.

For the accuracy metric, we adjusted the objective function as reg:squared with rmse as the evaluation metric, the maximum tree depth to 5, the learning rate to 0.01, and the number of estimators to 300, while keeping other hyperparameters fixed based on those originally established for the AUC evaluation.

Table 12: Accuracy results for heterophilic datasets with baselines from Li et al. (2022).

| Model | Texas | Wisconsin | Cornell |
|---|---|---|---|
| GCN | $55.14 \pm 5.16$ | $51.76 \pm 3.06$ | $60.54 \pm 5.30$ |
| GAT | $52.16 \pm 6.63$ | $49.41 \pm 4.09$ | $61.89 \pm 5.05$ |
| MixHop | $77.84 \pm 7.73$ | $75.88 \pm 4.90$ | $73.51 \pm 6.34$ |
| GPR-GNN | $78.38 \pm 4.36$ | $82.94 \pm 4.21$ | $80.27 \pm 8.11$ |
| LINKX | $74.60 \pm 8.37$ | $75.49 \pm 5.72$ | $77.84 \pm 5.81$ |
| GGCN | $\mathbf{84.86 \pm 4.55}$ | $86.86 \pm 3.29$ | $\mathbf{85.68 \pm 6.63}$ |
| GloGNN | $84.32 \pm 4.15$ | $\mathbf{87.06 \pm 3.53}$ | $83.51 \pm 4.26$ |
| Ours | $73.98 \pm 3.45$ | $77.8 \pm 2.67$ | $79.11 \pm 3.37$ |

