# OpenReview forum: "Bag of Features: New Baselines for GNNs for Link Prediction"
_ICLR.cc/2024/Conference — Submitted to ICLR 2024_

### Official Review · Reviewer_in5m · 2023-10-16

**Soundness:** 3 good
**Presentation:** 3 good
**Contribution:** 4 excellent
**Rating:** 8
**Confidence:** 4

**Summary:**

This paper investigates the performance of Graph Neural Networks (GNNs) for link prediction tasks in comparison to traditional feature extraction methods. The authors propose a simple machine learning model called "Bag of Features," which combines structural features based on node proximity in the graph structure and domain features based on feature space similarity. Their findings show that this model delivers highly competitive results when compared to state-of-the-art GNN models across various benchmark datasets. This supports recent theoretical observations that current GNNs may not be fully exploiting their potential, suggesting a need for innovative approaches to unlock the full capabilities of GNNs.

**Strengths:**

1. The proposed Bag of Features (BFLP) model is elegant, simple, and effective, combining both structural features and domain features to create a powerful machine learning model for link prediction tasks.

2. The model demonstrates consistently competitive results when compared to state-of-the-art GNN models, providing empirical evidence that there is room for improvement in current GNN approaches.

3. The paper highlights the potential for conventional feature extraction methods to be integrated into future ML models to enhance their performance.

4. The work is likely to inspire future research in the link prediction domain, encouraging researchers to explore alternative methods and improve upon existing GNN techniques.

Overall, I enjoyed reading the paper and it is a significant contribution to the application of link prediction.

**Weaknesses:**

Several improvements can be made to enhance the quality of the work. Addressing these comments may lead to a higher score:
1. Rectify the discrepancies between baseline performances in Table 5 and those on the OGB website (refer to Comment 1 for details).

2. Broaden the evaluation datasets to encompass a variety of network structures and domains (see Comments 2 and 3 for clarification).

3. Perform ablation studies to gain a deeper understanding of the contributions made by the Bag of Features model components (refer to Comment 4 for more information).

**Questions:**

1. A discrepancy exists between the performance reported in Table 5 and the OGB website. For instance, GraphSAGE shows a 48.10 on the OGB website, but the paper reports 56.88. The performance in Table 5 closely aligns with the validation performance on the OGB website. The authors should verify if Table 5 reflects the validation dataset performance and determine the cause of this inconsistency.

2. Tables 3 and 4 follow the methods of (Zhao et al., 2022) and (Guo et al., 2022), respectively. To enhance comprehensiveness, the authors should consider adding Facebook and OGB-DDI to Table 3, and Cora, Citeseer, and DBLP to Table 4.

3. The paper focuses on homophilous graph datasets, but the proposed method could also be applied to heterophyllous graph datasets. Including these datasets would increase the paper's impact.

4. The ablation studies only examine the removal of all structural or domain features. A more detailed table illustrating the impact of each individual feature on the datasets would make the results more appealing.

---

> ### Author Response · Authors · 2023-11-12
> **Responses to Reviewer in5m**
>
> We express our gratitude to the reviewer for the insightful and constructive feedback. In light of your valuable comments, as well as those from other reviewers, we are revising our paper. We are confident that these revisions have substantially enhanced the quality of our work. We hope that our responses will be sufficiently informative for you to reconsider assessing a higher rating for the revised paper.
>
> $\textbf{Q1. Table Discrepancy.}$
>
> >Thanks for sharing this point. For OGB-Collab, we used the results given in [1],  a review paper that compares the performances of recent Graph Neural Networks (GNNs) in the link prediction task. All the reported performances are extracted from this source. On the OGB-Leaderboard, we observed that GraphSage's performance was suboptimal, so we opted for another configuration with improved hyperparameters.
>
> >In response to Reviewer Dape's request, we included additional GNN baselines in Table 5. These supplementary results are sourced from [1], specifically Table 2 in the reference.
>
> >[1] Li, J., Shomer, H., Mao, H., Zeng, S., Ma, Y., Shah, N., Tang, J. and Yin, D., 2023. Evaluating Graph Neural Networks for Link Prediction: Current Pitfalls and New Benchmarking. arXiv preprint arXiv:2306.10453.
>
> $\textbf{Q2. Extra Datasets:}$ Tables 3 and 4 follow the methods of (Zhao et al., 2022) and (Guo et al., 2022), respectively. To enhance comprehensiveness, the authors should consider adding Facebook and OGB-DDI to Table 3, and Cora, Citeseer, and DBLP to Table 4.
>
> >Thank you very much for this suggestion. These two papers use different splits (70/10/20 and 85/5/10) so we did not add our CORA/CITESEER results from Table 3 to Table 4 not to cause any confusion. We add these results to the appendix with different splits (Table 9). We will try to complete the experiments for the suggested datasets (FACEBOOK, DBLP, and OGB dataset) before the rebuttal deadline, and let you know the results.
>
> $\textbf{Q3. Heterophilic Datasets.}$
>
> >Thank you very much for this great suggestion. We will add experiments for some heterophilic datasets before the rebuttal deadline, and let you know the results.
>
> $\textbf{Q4. Ablation Study for Individual Features.}$
>
> >Thank you very much for this suggestion. We added the ablation study table for feature importance in the appendix (Table 7)
>
> Please let us know if you have further questions. Thank you very much again for your valuable time and feedback.

---

> > ### Author Response · Authors · 2023-11-22
> >
> > We express our gratitude to the reviewer for the insightful and constructive feedback. In light of your valuable comments, as well as those from other reviewers, we revised our paper. Here is the summary of revisions:
> >
> > 1. Feature Importance table (Table 7)
> > 2. Addition of new experiments for heterophilic datasets (Table 12)
> > 3. Addition of new ablation study for ML classifiers (Table 11).
> > 4. Addition of the latest GNN baselines on 5 datasets (Tables 8 and 9)
> > 5. New GNN baselines for OGB Collab (Table 5 and Table 10)
> >
> > Unfortunately, due to unforeseen issues, we could not finish OGBL-PPA experiments yet. However, we will add the results to our paper when finished. We would like to underline that our main goal and novelty in our study is not to advocate for the adoption of feature engineering methods moving forward. Instead, we aim to empirically document that current GNN models do not demonstrate a significant advantage over traditional methods as recent theoretical studies point out. This revelation should illuminate the current state of affairs with GNNs to the ML community and motivate researchers to advance and refine these models further.
> >
> > We are confident that these revisions have substantially enhanced the quality of our work. We hope that our responses will be sufficiently informative for you to reconsider assessing a higher rating for the revised paper. Thank you very much again for your valuable time and feedback.

---

> > > ### Comment · Reviewer_in5m · 2023-11-22
> > >
> > > Having had my concerns sufficiently addressed, I have decided to increase my rating. I regard this as a commendable paper within the application of link prediction (if the experimental setting in Table 5 is correct). In my view, the paper's significance doesn't lie in the proposed method, but rather in the call to rethink the feature learning of Graph Neural Networks (GNNs) in link prediction. This paper serves as an excellent springboard, and it is highly likely to inspire future advancements in the design of link prediction GNNs.

---

> > > > ### Author Response · Authors · 2023-11-22
> > > > **Thank you!**
> > > >
> > > > Dear Reviewer in5m,
> > > >
> > > > We are very grateful for your insightful feedback. We are pleased to note that our responses have effectively addressed your concerns. Your positive evaluation and recognition of our work mean a lot to us. Your time and insights in highlighting the contributions of this work are immensely valued.
> > > >
> > > > Best wishes!

---

### Official Review · Reviewer_Wcv5 · 2023-10-27

**Soundness:** 3 good
**Presentation:** 3 good
**Contribution:** 2 fair
**Rating:** 5
**Confidence:** 4

**Summary:**

This work unifies conventional feature extraction methods for link prediction and uses standard machine learning models to learn from them. The result model delivers highly competitive results on various datasets.

**Strengths:**

1. Clear method.

2. Solid experiments.

**Weaknesses:**

1. Novelty is limited. Most methods are ordinary feature extraction tricks.

2. Experiments are not conducted on OGB datasets other than collab.

**Questions:**

1. Please add experiments on other OGB datasets.

2. This work used XGBoost as the ML model. Did you conduct ablation study for it? For example, use Logistic regression and SVM instead?

---

> ### Author Response · Authors · 2023-11-12
> **Responses to Reviewer Wcv5**
>
> We express our gratitude to the reviewer for the insightful and constructive feedback. In light of your valuable comments, as well as those from other reviewers, we are revising our paper. We are confident that these revisions have substantially enhanced the quality of our work. We hope that our responses will be sufficiently informative for you to reconsider assessing a higher rating for the revised paper.
>
> $\textbf{W1. Novelty:}$ Novelty is limited. Most methods are ordinary feature extraction tricks.
>
> >Thank you very much for sharing this concern. We wish to emphasize that the novelty of our work lies in substantiating that traditional feature engineering methods can stand shoulder-to-shoulder with cutting-edge GNN models. We argue this based on consistently outperforming powerful GNN methods on various datasets, which are published in top conferences. This supports our claim that our Bag of Features model is a practical and effective alternative to these computationally intensive models.
>
> $\textbf{W2/Q1. Extra OGB experiments:}$ Experiments are not conducted on OGB datasets other than collab.
>
> >We are working on additional OGB experiments. We hope to provide the results before the rebuttal deadline.
>
> $\textbf{Q2. XGBoost vs. other ML classifiers:}$ This work used XGBoost as the ML model. Did you conduct ablation study for it? For example, use Logistic regression and SVM instead ?
>
> >Thank you very much for this suggestion. We added the ablation study table reporting the performance of several ML classifiers (including Naive Bayes and Logistic Regression)  in the appendix (Table 11). We will add other ML classifiers soon.
>
> Please let us know if you have further questions. Thank you very much again for your valuable time and feedback.

---

> > ### Author Response · Authors · 2023-11-22
> >
> > We express our gratitude to the reviewer for the insightful and constructive feedback. In light of your valuable comments, as well as those from other reviewers, we revised our paper. Here is the summary of revisions:
> >
> > 1. Feature Importance table (Table 7)
> > 2. Addition of new experiments for heterophilic datasets (Table 12)
> > 3. Addition of new ablation study for ML classifiers (Table 11).
> > 4. Addition of the latest GNN baselines on 5 datasets (Tables 8 and 9)
> > 5. New GNN baselines for OGB Collab (Table 5 and Table 10)
> >
> > Unfortunately, due to unforeseen issues, we could not finish OGBL-PPA experiments yet. However, we will add the results to our paper when finished. We would like to underline that our main goal and novelty in our study is not to advocate for the adoption of feature engineering methods moving forward. Instead, we aim to empirically document that current GNN models do not demonstrate a significant advantage over traditional methods as recent theoretical studies point out. This revelation should illuminate the current state of affairs with GNNs to the ML community and motivate researchers to advance and refine these models further.
> >
> > We are confident that these revisions have substantially enhanced the quality of our work. We hope that our responses will be sufficiently informative for you to reconsider assessing a higher rating for the revised paper. Thank you very much again for your valuable time and feedback.

---

> > > ### Comment · Reviewer_Wcv5 · 2023-11-22
> > >
> > > Thanks for the detailed response. However, results on other OGB dataset are still not available. Therefore, I would keep my score.

---

### Official Review · Reviewer_dape · 2023-11-01

**Soundness:** 2 fair
**Presentation:** 3 good
**Contribution:** 1 poor
**Rating:** 3
**Confidence:** 4

**Summary:**

This paper leverages different graph structural features with node attributes for link prediction tasks. Then, an XGBoost method is utilized based on these features for link prediction.

**Strengths:**

1. The authors explore different graph structural features for link prediction.
2. The paper is well-written and easy to follow.

**Weaknesses:**

1. The paper's novelty appears constrained. The significance of graph structural features in the realm of link prediction has been previously underscored by several studies, such as [1][2][3].
2. The evaluation omits several critical baselines that similarly exploit graph structural features. Notably absent are BUDDY, Neo-GNN, NCN, NCNC, and NBFNet.
3. The experimental setup seems to miss out on key datasets such as ogbl-ddi, ogbl-ppa, and ogbl-citation2.
4. There are no ablation studies about the importance of each feature.
5. While the method demonstrates commendable efficacy on the OGBL-Collab dataset, it's concerning that the associated code is inaccessible, given the empty repository link.
6. Using the original feature to measure the node similarity might be unreasonable.

[1] Menon, Aditya Krishna, and Charles Elkan. "Link prediction via matrix factorization."
[2] Muhan Zhang and Yixin Chen. Link prediction based on graph neural networks.
[3] Xiyuan Wang, Haotong Yang, and Muhan Zhang. Neural common neighbor with completion
for link prediction.

**Questions:**

Please refer to the weaknesses.

---

> ### Author Response · Authors · 2023-11-12
> **Responses to Reviewer dape**
>
> We express our gratitude to the reviewer for the insightful and constructive feedback. In light of your valuable comments, as well as those from other reviewers, we are revising our paper. We are confident that these revisions have substantially enhanced the quality of our work. We hope that our responses will be sufficiently informative for you to reconsider assessing a higher rating for the revised paper.
>
> $\textbf{W1: Novelty:}$ The paper's novelty appears constrained. The significance of graph structural features in the realm of link prediction has been previously underscored by several studies, such as [1][2][3].
>
> >Thank you very much for bringing these references to our attention. We added these references to our paper. We wish to emphasize that the novelty of our work lies in substantiating that traditional feature engineering methods can stand shoulder-to-shoulder with cutting-edge GNN models. We argue this based on consistently outperforming powerful GNN methods on various datasets, which are published in top conferences. This supports our claim that our Bag of Features model is a practical and effective alternative to these computationally intensive models.
>
> $\textbf{W2. Sota Baselines:}$ The evaluation omits several critical baselines that similarly exploit graph structural features. Notably absent are BUDDY, Neo-GNN, NCN, NCNC, and NBFNet.
>
> >By using reference [1], we give the performance comparison of these GNN models with our model. For OGBL-COLLAB, we report Hits@50 results below, and add them into our accuracy table (Table 5). However, for CORA, CITESEER, and PUMED datasets, the reported performances use different splits. Considering our model uses less training data, it produces highly competitive results with these cutting edge GNN models. Since the experimental setups are different, we opt out these performances  in our accuracy table in the main text, however we added this table to appendix (Table 7).
>
> >OGBL-COLLAB
> | Model   | Hits@50      |
> |---------|--------------|
> | SEAL    | 63.37 ± 0.69 |
> | BUDDY   | 64.59 ± 0.46 |
> | Neo-GNN | 66.13 ± 0.61 |
> | NCN     | 63.86 ± 0.51 |
> | NCNC    | 65.97 ± 1.03 |
> | NBFNet  | OOM          |
> | Ours    | 76.50 ± 0.27 |
>
> >| Model    | Split      | CORA         | CITESEER     | PUBMED       |
> |----------|------------|--------------|--------------|--------------|
> | BUDDY    | 85:05:10    | 95.06±0.36   | 96.72±0.26   | 98.20±0.05   |
> | Neo-GNN  | 85:05:10    | 93.73±0.36   | 94.89±0.60   | 98.71±0.05   |
> | NCN      | 85:05:10    | 96.76±0.18   | 97.04±0.26   | 98.98±0.04   |
> | NCNC     | 85:05:10    | 96.90±0.28   | 97.65±0.30   | 99.14±0.03   |
> | NBFNet   | 85:05:10    | 92.85±0.17   | 91.06±0.15   | 98.34±0.02   |
> | Ours     | 70:10:20    | 94.54±0.30   | 95.01±0.35   | 94.73±0.15   |
>
> >[1] Li, J., Shomer, H., Mao, H., Zeng, S., Ma, Y., Shah, N., Tang, J. and Yin, D., 2023. Evaluating Graph Neural Networks for Link Prediction: Current Pitfalls and New Benchmarking. arXiv preprint arXiv:2306.10453.
>
> $\textbf{W3. Extra OGB experiments:}$
>
> >We are working on additional OGB experiments. We hope to provide the results before the rebuttal deadline. We will let you know when we have the results.
>
> $\textbf{W4. Ablation Study.}$ There are no ablation studies about the importance of each feature.
>
> >Thank you very much for this suggestion. We added the ablation study table reporting the impact of each individual feature in the appendix (Table 7), which you can see in our revision.
>
> $\textbf{W5. No Code:}$
>
> >Thanks for letting us know about the mistake in the link. We've fixed it in the paper, and you can find the correct link below. If you have any more questions about our code, please let us know.
>  https://github.com/workrep20232/LinkPrediction
>
> $\textbf{W6. Using original features for node similarity?}$ Using the original feature to measure the node similarity might be unreasonable.
> >Thanks for raising this concern. Our primary objective with our approach is to provide a highly generalized method for gathering all relevant similarity/dissimilarity information between node pairs and leveraging these features in the task of link prediction. You correctly point out that the efficacy of domain features may vary depending on the dataset's nature, whether it exhibits homophily or heterophily. We acknowledge that, in the PHOTO and COMPUTERS datasets, domain features did not contribute substantially to performance. In contrast, for CORA, CITESEER, and PUBMED datasets, the inclusion of domain features notably enhanced performance. It's noteworthy that incorporating these features didn't compromise performance across all datasets; instead, it consistently improved performance across the board. You can check our Feature Importance table (Table 7) in the appendix.
>
> Please let us know if you have further questions. Thank you very much again for your valuable time and feedback.

---

> > ### Comment · Reviewer_dape · 2023-11-17
> >
> > Thanks for the response, which has addressed some of my concerns. However, I still have some concerns:
> > 1. The data splits of the proposed method and baselines in W2 are different. As a result, it's not easy to make a fair comparison.
> > 2. How do you address the scalability issue? It seems the time complexity is quite high.
> > 3. Waiting for the results of OGB datasets.

---

> ### Author Response · Authors · 2023-11-22
>
> Thank you very much for your questions.
>
> **1. Different data splits.**
>
> Yes, in our original submission, we used the performances reported in two very recent papers' results as baselines [1,2], but they were using different data splits. So, we had one table for Cora/Citeseer/PubMed (70:10:20), and Photos/Computers (85:5:10). However, with the request of Reviewers rEua and in5m, we added these comparisons with different splits for Cora/Citeseer/PubMed. In our original submission, we did not want to cause any confusion by giving two different results for the same datasets. To avoid the confusion, we give these results in the appendix. If needed, we can repeat the experiments with new splits, and report them in the appendix.
>
> **2. Scalability Issue**
>
> We believe scalability is one of the advantages of our simple approach. We give the details of our runtime and computational complexities in Experiments section. The computational complexities of our similarity indices are $\mathcal{O}(|V|.k^3)$ where $|V|$ is the total number of nodes and $k$ is the maximum degree in the network. For a large network, the computational time needed for our approach is much less than the training time for a message passing GNN. Therefore, our approach provides a good alternative to common methods when dealing with large networks.
>
> **3. OGB Datasets.**
>
> Unfortunately, due to unforeseen circumstances, we could not complete OGBL-PPA experiments yet. We will add the results to the paper when we have it. However, in OGBL-COLLAB, we outperform all the existing results with our simple model. We are expecting a competitive result with OGB Leaderboard in OGBL-PPA dataset considering its high performance in all other (both homophilic and heterophilic) datasets, too.
>
> Thank you very much for your valuable time and feedback. Please let us know if you have further questions.
>
> [1] Li, Juanhui, Harry Shomer, Haitao Mao, Shenglai Zeng, Yao Ma, Neil Shah, Jiliang Tang, and Dawei Yin. "Evaluating Graph Neural Networks for Link Prediction: Current Pitfalls and New Benchmarking." arXiv preprint arXiv:2306.10453 (2023).
>
> [2] Guo, Zhihao, Feng Wang, Kaixuan Yao, Jiye Liang, and Zhiqiang Wang. "Multi-scale variational graph autoencoder for link prediction." In Proceedings of the Fifteenth ACM International Conference on Web Search and Data Mining, pp. 334-342. 2022.

---

> > ### Author Response · Authors · 2023-11-22
> >
> > We express our gratitude to the reviewer for the insightful and constructive feedback. In light of your valuable comments, as well as those from other reviewers, we revised our paper. Here is the summary of revisions:
> >
> > 1. Feature Importance table (Table 7)
> > 2. Addition of new experiments for heterophilic datasets (Table 12)
> > 3. Addition of new ablation study for ML classifiers (Table 11).
> > 4. Addition of the latest GNN baselines on 5 datasets (Tables 8 and 9)
> > 5. New GNN baselines for OGB Collab (Table 5 and Table 10)
> >
> > Unfortunately, due to unforeseen issues, we could not finish OGBL-PPA experiments yet. However, we will add the results to our paper when finished. We would like to underline that our main goal and novelty in our study is not to advocate for the adoption of feature engineering methods moving forward. Instead, we aim to empirically document that current GNN models do not demonstrate a significant advantage over traditional methods as recent theoretical studies point out. This revelation should illuminate the current state of affairs with GNNs to the ML community and motivate researchers to advance and refine these models further.
> >
> > We are confident that these revisions have substantially enhanced the quality of our work. We hope that our responses will be sufficiently informative for you to reconsider assessing a higher rating for the revised paper. Thank you very much again for your valuable time and feedback.

---

> > ### Comment · Reviewer_dape · 2023-11-22
> >
> > Thanks for the further response. However, it doesn't solve my question. First, you should use the same data split for your method and the baselines to make a fair comparison. Second, I suggest three ogb datasets, i.e., ogbl-ddi, ogbl-ppa, and ogbl-citation2, but I don't see any results about these three datasets.

---

> ### Author Response · Authors · 2023-11-22
>
> **1. Different Splits**
>
> We want to clarify that our Table 3 is all 70:10:20 using baselines from (Guo et all 2022). We are not using different split there.
>
> | Models   | CORA       | CITESEER   | PUBMED     |
> |----------|------------|------------|------------|
> | Node2Vec | 84.49 ± 0.49   | 80.00 ± 0.68       | 80.32 ± 0.29     |
> | MVGRL    | 75.07 ± 3.63   | 61.20 ± 0.55       | 80.78 ± 1.28     |
> | VGAE     | 88.68 ± 0.40   | 85.35 ± 0.60       | 95.80 ± 0.13     |
> | SEAL     | 92.55 ± 0.50   | 85.82 ± 0.44       | 96.36 ± 0.28     |
> | GCN      | 90.25 ± 0.53   | 71.47 ± 1.40       | 96.33 ± 0.80     |
> | GSAGE    | 90.24 ± 0.34   | 87.38 ± 1.39       | 96.78 ± 0.11     |
> | JKNet    | 89.05 ± 0.67   | 88.58 ± 1.78       | 96.58 ± 0.23     |
> | CFLP     | 93.05 ± 0.24   | 92.12 ± 0.47       | **97.53 ± 0.17**     |
> | Ours| **94.54 ± 0.30**   | **95.01 ± 0.35**       | 94.73 ± 0.15     |
>
> For Tables 8 and 9, we are adding new GNN baselines by using the performances from (Li et al, 2023) which uses 85:05:10 split. If your question is about these two tables, we are sorry that we misunderstood your concern.  We will add the experiments with 85:05:10 splits for Cora/Citeseer/PubMed datasets. Since we already have the features, it should take only a couple of hours. We'll get back to you soon.
>
> **2. OGB Datasets**
>
> Unfortunately, we could not finish the OGB experiments in time. Considering the size of these datasets, we had to use our university clusters. We were expecting to finish one of these datasets in time: OGBL-PPA. We were able to get the features, however, we had a problem with running ML classifiers.
>
> Nonetheless, we were not expecting to finish all three OGB datasets in one week considering their sizes. Our structural features are always the same, but domain features need to be customized and it should be tailored for each dataset. For PPA, we were able to do it, but since OGBL-Citation2 and OGBl-DDI are completely different formats, we need time to extract meaningful features from the node and edge features provided.
>
> Again, we want to underline that our aim is not to promote feature engineering models but to show that current GNN performances are not far from traditional methods, verifying the recent theoretical studies. Thank you very much for your time and feedback.

---

> > ### Author Response · Authors · 2023-11-22
> > **Different splits**
> >
> > Dear Reviewer dape,
> >
> > Here are our results with 85:05:10 split as promised. The baselines in the table below are all in the same setting.
> >
> > Note that in limited time, we could not do any hyperparameter tuning, we just used the same hyperparameters in XGBoost from 70:10:20 setting. We believe with further tuning, we could improve these results. We will update our results in the appendix after hyperparameter tuning. The baseline details can be found in Appendix B.2.
> >
> > Please let us know if you have further questions or remarks. Thank you very much again for your valuable time and feedback.
> >
> > | Model    | CORA        | CITESEER    | PUBMED      |
> > |----------|-------------|-------------|-------------|
> > | GAE      | 91.5±0.02    | 91.0±0.04    | --           |
> > | VGAE     | 91.5±0.04    | 91.2±0.03    | --           |
> > | ARGA     | 91.5±0.03    | 92.8±0.03    | --           |
> > | ARVGA    | 93.1±0.03    | 92.5±0.01    | --           |
> > | DBGAN    | 94.5±0.01    | 94.5±0.04    | --           |
> > | MSVGAE   | 95.3±0.05    | 95.4±0.03    | --           |
> > | BUDDY    | 95.06±0.36  | 96.72±0.26  | 98.20±0.05  |
> > | Neo-GNN  | 93.73±0.36  | 94.89±0.60  | 98.71±0.05  |
> > | NCN      | 96.76±0.18  | 97.04±0.26  | 98.98±0.04  |
> > | NCNC     | **96.90±0.28**  | **97.65±0.30**  | **99.14±0.03**  |
> > | NBFNet   | 92.85±0.17  | 91.06±0.15  | 98.34±0.02  |
> > |----------|----------|-------------|-------------|-------------|
> > | Ours     | 95.01±0.32  | 95.95 ±0.38  | 96.26 ±0.13  |

---

> > > ### Comment · Reviewer_dape · 2023-11-22
> > >
> > > Dear Authors,
> > >
> > > Thanks for the efforts. The fact that structural features are useful for link prediction is a well-known fact. Thus, state-of-the-art methods, such as NCNC, leverage these features can improve the performance of GNNs. From the results in your rebuttal, we can see that just leveraging all the features cannot outperform GNNs, although it can achieve relative good performance. In my opinion, just verifying this phenomenon is not sufficient for an ICLR paper. I recommend the authors go beyond this verification and leverage these features to design some new methods.
> > >
> > > Best Regards,
> > >
> > > Reviewer dape

---

> > > > ### Author Response · Authors · 2023-11-22
> > > > **Emperor's New Clothes**
> > > >
> > > > Dear Reviewer dape,
> > > >
> > > > Thank you very much for sharing your insights. As we mentioned in our discussion at the end, we already started working on leveraging the insights we got from this study to develop novel GNN models for link prediction and node classification.
> > > >
> > > > One challenge with GNNs is that they initiate with node embeddings and try to refine these embeddings with each epoch, guided by the loss function's negative gradient. In numerous instances, if there's a node feature vector available, it's incorporated into the initial node embedding, while the graph proximity information is indirectly introduced during the message passing phase. As these node embeddings are refined over iterations, the model employs them as needed, depending on the task (e.g., node classification, graph classification, link prediction).
> > > >
> > > > Firstly, methods like ours, focused on feature extraction, can be instrumental in the conventional GNN framework in the initial embedding phase. It's essential to note that node embeddings undergo updates after every iteration. Thus, beginning with robust node embeddings like ours can significantly enhance performance. Consequently, our feature vectors could serve as the initial embeddings, considering they also capture graph structural data. Given that our simple ML model competes well with numerous GNN models, incorporating our features should give them a performance edge. We are exploring this avenue as a follow-up research. This is only one approach, and we are exploring other directions forward.
> > > >
> > > > While we value your perspective, we believe it is not that obvious that an adaptation of conventional methods can compete with SOTA GNNs in both homophilic and heterophilic datasets in link prediction tasks. We believe that our paper highlights this surprising fact in a manner reminiscent of the tale of *Emperor's new clothes*, and provides a valuable contribution to the ML community by empirically showing the current state of affairs.
> > > >
> > > > Thank you very much again for your valuable time.

---

### Official Review · Reviewer_rEua · 2023-11-02

**Soundness:** 2 fair
**Presentation:** 2 fair
**Contribution:** 2 fair
**Rating:** 3
**Confidence:** 4

**Summary:**

This paper proposes bags of heuristics for link prediction. The experiments show the effectiveness of the proposed method.

**Strengths:**

This paper shows that the conventional features of the graph can be utilized to outperform GNN in link prediction.

**Weaknesses:**

1. The conclusion that graph heuristics can outperform GNN in link prediction is not a new contribution. This article only proposes some heuristics but does not explain why they are effective.
2. This paper overclaims its contribution. The method and experiments of this paper are designed for link prediction, but the contribution of this paper lies in discussing that GNN is not good at graph learning. As far as I know, heuristics perform better than GNN only in link prediction, and there seems to be no similar phenomenon in other domains such as graph classification, node classification, and graph regression.
3. The experimental part of this article does not compare with state-of-the-art GNN methods, such as NBFNet[1] and Seal[2].

[1] Neural Bellman-Ford Networks: A General Graph Neural Network Framework for Link Prediction

[2] Link Prediction Based on Graph Neural Networks

**Questions:**

see weakness

---

> ### Author Response · Authors · 2023-11-12
> **Responses to Reviewer rEua**
>
> We express our gratitude to the reviewer for the insightful and constructive feedback. In light of your valuable comments, as well as those from other reviewers, we are revising our paper. We are confident that these revisions have substantially enhanced the quality of our work. We hope that our responses will be sufficiently informative for you to reconsider assessing a higher rating for the revised paper.
>
> $\textbf{W1\ - \ Novelty:}$
>
> >Thank you very much for sharing this concern. We wish to emphasize that the novelty of our work lies in substantiating that traditional feature engineering methods can stand shoulder-to-shoulder with cutting-edge GNN models. We argue this based on consistently giving shoulder-to-shoulder performance with powerful GNN methods on various datasets, which are published in top conferences. This supports our claim that our Bag of Features model is a scalable and effective alternative to these computationally intensive models.
>
> >The key reason behind our success is revealed in our ablation study. Our approach of effectively feeding both structural and domain features in our ML classifier has proven to be crucial. The study confirms that, in most datasets, using only structural or domain features leads to lower performance. However, combining them effectively significantly boosts the overall performance of our model.
>
> $\textbf{W2. GNN comparison for Other Graph Learning Tasks}$
>
> >Thank you for raising this issue. It's accurate to observe that our focus in this manuscript centers specifically on the challenge of link prediction. The decision to narrow our attention to this particular task within graph representation learning was primarily influenced by space limitations. In a distinct investigation, we employed similar techniques to compare Graph Neural Networks (GNNs) with traditional feature extraction in the context of a node classification task. Across six benchmark datasets in that study, we outperformed all state-of-the-art GNNs. While our findings with the OGBGN dataset did not outperform the OGB leaders, but gave highly competitive results. To preserve anonymity, we refrain from providing additional details.
>
> >Given the inherent differences between node classification and link prediction tasks, the employed feature extraction methods also diverge significantly. Consequently, we opted to present these two studies in separate papers. Nevertheless, the outcomes from both studies imply that existing GNNs do not exhibit a substantial superiority over traditional feature engineering models. It is crucial to clarify that our intention in this paper is not to advocate for the adoption of feature engineering methods moving forward. Instead, we aim to document that current GNN models do not demonstrate a significant advantage over them. This revelation should illuminate the current state of affairs with GNNs and motivate researchers to advance and refine these models further.
>
> $\textbf{W3. SOTA Baselines:}$ The experimental part of this article does not compare with state-of-the-art GNN methods, such as NBFNet[1] and Seal[2].
>
> >Thanks for pointing out these baseline references. It's important to mention that SEAL [2] has already been given in Table 3 - row 4. For NBFNet, we report the result for a different split of 85:5:10 below by using reference [3]. However, since we use a different split for Table 3 (70:10:20), we opted not to include NBFNet in Table 3. On the other hand, with the suggestion of Reviewer Dape, we added several recent GNN baselines to our Table 5, and Table 8. We also acknowledged this reference in our related work section.
>
> >The specifics of AUC performances are outlined in the table provided below.
> |      | Split      | CORA  | CITESEER | PUBMED |
> |------|------------|-------|----------|--------|
> | SEAL | 70:10:20    | 92.55 | 85.82    | 96.36  |
> | NBFNet | 85:05:10  | 95.60  | 92.30     | 98.30   |
> | Ours | 70:10:20    | 94.54 | 95.01    | 94.73  |
>
> >[3] Li, J., Shomer, H., Mao, H., Zeng, S., Ma, Y., Shah, N., Tang, J. and Yin, D., 2023. Evaluating Graph Neural Networks for Link Prediction: Current Pitfalls and New Benchmarking. arXiv preprint arXiv:2306.10453.
>
> Please let us know if you have further questions. Thank you very much again for your valuable time and feedback.

---

> > ### Author Response · Authors · 2023-11-22
> >
> > We express our gratitude to the reviewer for the insightful and constructive feedback. In light of your valuable comments, as well as those from other reviewers, we revised our paper. Here is the summary of revisions:
> >
> > 1. Feature Importance table (Table 7)
> > 2. Addition of new experiments for heterophilic datasets (Table 12)
> > 3. Addition of new ablation study for ML classifiers (Table 11).
> > 4. Addition of the latest GNN baselines on 5 datasets (Tables 8 and 9)
> > 5. New GNN baselines for OGB Collab (Table 5 and Table 10)
> >
> > Unfortunately, due to unforeseen issues, we could not finish OGBL-PPA experiments yet. However, we will add the results to our paper when finished. We would like to underline that our main goal and novelty in our study is not to advocate for the adoption of feature engineering methods moving forward. Instead, we aim to empirically document that current GNN models do not demonstrate a significant advantage over traditional methods as recent theoretical studies point out. This revelation should illuminate the current state of affairs with GNNs to the ML community and motivate researchers to advance and refine these models further.
> >
> > We are confident that these revisions have substantially enhanced the quality of our work. We hope that our responses will be sufficiently informative for you to reconsider assessing a higher rating for the revised paper. Thank you very much again for your valuable time and feedback.

---

### Author Response · Authors · 2023-11-12
**Summary of Revisions**

Dear Reviewers,

We are very grateful to all the reviewers for their insightful and constructive feedback. In light of your valuable comments, we completed the initial revisions of the paper. The summary of the current revisions:

1. Feature Importance table (Table 7)
2. Addition of the latest GNN baselines on 5 datasets (Tables 8 and 9)
3. New GNN baselines for OGB Collab (Table 5 and Table 10)
4. Ablation study for ML classifiers (Table 11).

We are currently working on new experiments with our model for OGB and heterophilic datasets. We hope to finish them before the rebuttal deadline and keep you posted.

We provided our responses to your questions below. *We would like to underline that our main goal and novelty in our study is not to advocate for the adoption of feature engineering methods moving forward. Instead, we aim to empirically document that current GNN models do not demonstrate a significant advantage over traditional methods as recent theoretical studies point out. This revelation should illuminate the current state of affairs with GNNs and motivate researchers to advance and refine these models further.*

Please let us know if you have further questions. Thank you very much again for your valuable time and feedback.

---

### Author Response · Authors · 2023-11-21
**New Experiment Results**

Dear Reviewers:

Here are the results for our heterophilic datasets. The baselines are taken from ICML 2022 paper [1].

| Model    | Texas        | Wisconsin    | Cornell      |
|----------|--------------|--------------|--------------|
| GCN      | 55.14±5.16   | 51.76±3.06   | 60.54±5.30   |
| GAT      | 52.16±6.63   | 49.41±4.09   | 61.89±5.05   |
| MixHop   | 77.84±7.73   | 75.88±4.90   | 73.51±6.34   |
| GPR-GNN  | 78.38±4.36   | 82.94±4.21   | 80.27±8.11   |
| LINKX    | 74.60±8.37   | 75.49±5.72   | 77.84±5.81   |
| GGCN     | **84.86±4.55**   | 86.86±3.29   |**85.68±6.63**   |
| GloGNN   | 84.32±4.15   | **87.06±3.53**   | 83.51±4.26   |
| Ours     | *73.98±3.45*   | *77.82±2.67*   | *79.11±3.37*   |

We'd like to highlight that our model isn't tailored specifically for heterophilic datasets; rather, it's designed to effectively handle both homophilic and heterophilic datasets. While our model demonstrates superior performance compared to standard GNN models, it might not match the performance of GNN models specifically optimized for this particular setting. We added this table and hyperparameter settings in Appendix B.4 (Table 12).

For OGBL-PPA experiments, we successfully extracted the features, however, because of a general problem in our clusters, we could not finish the experiments. Because of the holiday season, we could not reach our university staff to resolve the issue. We hope to finish the experiments by tomorrow. We thank you for your time and patience.

[1] Li, Xiang, Renyu Zhu, Yao Cheng, Caihua Shan, Siqiang Luo, Dongsheng Li, and Weining Qian. "Finding global homophily in graph neural networks when meeting heterophily." In International Conference on Machine Learning, pp. 13242-13256. PMLR, 2022.

---

### Meta-Review · Area_Chair_i2A9 · 2023-12-09

**Metareview:**

The paper shows standard ML methods over structural features can outperform state-of-the-art GNN link prediction models on some datasets.  Firstly, the datasets used are limited, with several other important benchmarks in OGB missing as pointed out by multiple reviewers. Second, the phenomenon itself is not new. Thus, the contribution is limited. I suggest the authors to investigate instead on how to improve the GNN expressivity for link prediction to let it automatically learn such features. Otherwise, the paper is more like a experimental report with marginal technical contribution.

**Justification For Why Not Higher Score:**

Limited technical contribution.

**Justification For Why Not Lower Score:**

N/A

---

### Decision · Program_Chairs · 2024-01-16

Reject